# A Network Perspective on the Relationship between Screen Time, Executive Function, and Fundamental Motor Skills among Preschoolers

**DOI:** 10.3390/ijerph17238861

**Published:** 2020-11-28

**Authors:** Clarice Maria de Lucena Martins, Paulo Felipe Ribeiro Bandeira, Natália Batista Albuquerque Goulart Lemos, Thaynã Alves Bezerra, Cain Craig Truman Clark, Jorge Mota, Michael Joseph Duncan

**Affiliations:** 1Department of Physical Education, Federal University of Paraiba, João Pessoa-PB 58000-000, Brazil; claricemartinsufpb@gmail.com (C.M.d.L.M.); thaynaalves.ef@gmail.com (T.A.B.); 2Department of Physical Education, Universidade Regional do Cariri—URCA; Crato-CE 63105-000, Brazil; paulo.bandeira@urca.br; 3Department of Physical Education, Federal University of Vale do São Francisco, Petrolina-PE 56304917, Brazil; 4Faculty of Health and Life Sciences, Coventry University, Priory Street, Coventry CV1 5FB, UK; ad0183@coventry.ac.uk (C.C.T.C.); aa8396@coventry.ac.uk (M.J.D.); 5Centre of Physical Activity, Health and Leisure, Faculty of Sport Sciences, University of Porto, 4500 Porto, Portugal; jmota@fade.up.pt

**Keywords:** screen time, fundamental motor skills, executive function, network perspective

## Abstract

The present study aimed to analyze the dynamic and nonlinear association between screen time, executive function (EF), and fundamental motor skills (FMS) in preschoolers, considering sex and body mass index (BMI) from a network perspective. Forty-two preschoolers (24 boys, 3.91 ± 0.77 years old) provided screen time, EF, FMS, and BMI data. EF was measured using the Go/No Go task, and accuracy of Go (sustain attention), reaction time of Go, and accuracy of No Go (inhibitory control) were considered. Relationships between screen time, EF, FMS, sex, and BMI were explored using a network analysis. The emerged network highlights that screen time is intensely associated with the other variables in the network, while the accuracy of Go has the greater connectivity with other nodes in the network (2.27), being the most sensitive to potential intervention changes. Moreover, sex (1.74), screen time (0.93), and accuracy of Go (0.71) showed the greatest closeness. This study showed that in the emerged network, independent of sex, screen exposure affects the accuracy on Go task, and these components affect the variables in the network, as motor abilities and tasks involved in inhibitory control.

## 1. Introduction

Executive function (EF) is a general term referring to complex cognitive processes needed for performing challenging goal-directed tasks, especially those that escape routine, and is composed of three core elements: inhibitory control (IC), working memory (WM), and cognitive flexibility (CF) [1,2,3]. These EF elements are based in the prefrontal cortex, and considered the basis of higher order cognitive functions, emerging with the brain and neural network, during the rapid stages of brain development [4,5]. However, studies drew attention to the impurity of EF tests in childhood, since the performance of young children in EF tasks seems to represent a confluence of cognitive processes of EF and non-EF [6,7].

Previous studies have widely assessed the IC as a measure of EF in this population, considering the preschool period has been identified as the key in its emergence [8]. Thus, EF is of extreme importance during the first years of life, as children who start school with low IC have difficulty in developing responses, and paying attention [9]. Even so, different measures of inhibition tasks, such as reaction time on accuracy, and accuracy on inhibition have been neglected. More recently, the computerized EF assessments have allowed researchers to summarize individual’s performance both on accuracy and reaction time (RT) data [10]. 

In the last few years, studies have given greater attention to the relationship between EF and children’s motor development, due to the close connection between brain development and motor abilities [11]. Indeed, higher-order cognitive processes and motor performance are closely linked, as they occur in the same organism, over the same period, and share the same brain basis [11,12]. Stöckel and Hughes [13] showed that in typically developing children, motor skills performance was positively related to EF. Roebers and Kauer [14] reported similar results for the association between inhibition measures and whole-body coordination in children, after controlling for age [14]. Although research has indicated the positive relationship between fundamental motor skills (FMS) and EF, environmental and behavioral factors, such as parental relationships [15], income status [16], and movement behaviors [17] as time exposed being sedentary is also related to EF, it may interfere in the association between EF and FMS. It is therefore important to consider this possibility.

Recent studies have shown that a considerable number of preschoolers worldwide are not compliant with screen time [18], and are excessively exposed to screens since the first year of life [19,20,21]. The rise of screen-based media use in preschool years may pose concerns regarding children’s neurobiological development, affecting how children play, learn, and form relationships [22,23]. A recent cross-sectional study with 3–5 year-old preschoolers showed that screen use greater than the recommended was associated with lower microstructural organization and myelination of brain white matter tracts [24]. Several cross-sectional and longitudinal studies support the notion that screen exposure may be associated with deleterious effects on children’s cognitive abilities [25,26]. Nonetheless, for preschool-aged children, the association between screen time and EF is controversial, due to the type of programming, and the social context of viewing. Moreover, preschoolers who spend excessive time in sedentary behaviors are more likely to have lower FMS scores [27]. 

Although often examined separately, the relationship between screen exposure, FMS, and EF in preschoolers could be better understood through the science of networks. EF is derived from complex neural connections [28,29], and is influenced by interactions with the environment. For example, with screen time. In this sense, methodologies based on complexity may be useful to understand the dynamic interaction of cognitive processes with related variables, such as FMS and screen time, allowing to assess the role of each variable within a complex, interconnected, and dynamic system. Network science, as a way of measuring complex systems, provides a topological structure of a network that can be useful to understand nonlinear relationships and is very sensitive to initial conditions. In addition, the role of each variable in the network can be better understood from the centrality measures, used to identify critical areas in the network that may be optimized though intervention processes [30]. Network science is an emerging research area used to understand and to measure complex systems. Therefore, this study aimed to analyze the dynamic and nonlinear association between compliance with screen time recommendation, EF, and FMS in preschoolers. 

## 2. Methods

### 2.1. Study Description

This cross-sectional study used baseline data from the “Movement’s Cool” project, aiming to analyze the association between physical activity (PA) and health outcomes in preschoolers. All the ethical aspects were followed. The evaluation methods and procedures were approved by the Research Ethics Committee of Health Science Center (protocol n. 2,727,698), and by the Education Board of the João Pessoa, Brazil city.

### 2.2. Participants and Context

Preschool children, aged 3–5 years old, of both sexes, and registered in Early Education Childhood Centers (EECC) of João Pessoa/Brazil were eligible. The preschool public education zone is organized in nine districts, where the 55 EECC 3–5 years old registered students are located. Ten institutions, located in deprived areas of six different districts, agreed to participate in the study. For this study, three EECC, situated in three different districts, were randomly selected. The three preschools were located in deprived areas, with low socioeconomic status (SES): 50.5% of the mothers or fathers were unemployed, and over 71.8% of the mothers had not finished high school. The Human Development Index of the EECC areas ranges from 0.4 to 0.5. A total of 42 children completed the entire assessment protocols and composed the final sample.

### 2.3. Study Design

Measurements were performed over four months (August to October 2019 and March 2020). All the schools and parents were informed about the project’s protocols and procedures in meetings with the project coordinator, and agreed to participate. All children authorized by their parents were evaluated.

A prior meeting with the school’s manager was conducted during the first day at school. On the second day, the sociodemographic data and screen time were provided by parents, and the Test of Gross Motor Development—Second Edition (TGMD-2) was applied. On the third day, EF data was collected.

### 2.4. Variables and Protocols

#### 2.4.1. Anthropometric Measures

Height (cm) and weight (kg) were determined using a Holtain stadiometer and digitized weighing scales (Seca 708), respectively, while the participant was lightly dressed and barefoot. BMI was calculated by dividing body weight with the squared height in meters (kg/m^2^).

#### 2.4.2. Screen Time

Parents were also asked to recall the total average duration their child watched TV, used the computer, and used videogames. The questions addressed weekdays and weekend days separately and were combined for analysis (Cronbach’s α = 0.87). For screen time the questions were: “How many hours during a week day does your child usually watch TV, use computer, smartphones, or electronics games?” and “How many hours during a weekend day does your child usually watch TV, use computer, smartphones, or electronics game?”. Then, the same procedure used for sleep hours was applied. Children were then classified as compliant/noncompliant with screen time when spending: (i) ≤1 h of sedentary screen time per day, for the 3 and 4 years-old children; (ii) ≤2 h of sedentary screen time per day for the 5 years-old children [31].

#### 2.4.3. Fundamental Motor Skills

FMS were measured using the TGMD-2 [32]. The TGMD-2 is valid and reliable for use in Brazilian children [33]. This test evaluates gross motor performance in children aged 3–10 years and consists of two factors: six locomotor skills (run, gallop, hop, leap, jump, and slide) and six ball skills (strike, bounce, catch, kick, throw, and underhand roll).

The TGMD-2 was administered at each preschool, according to the recommended guidelines [32]. Before the testing of each skill, participants were given a visual demonstration of the skill by the researcher using the correct technique but were not told what components of the skill were being assessed. Participants were then called individually to the practice trial. After that, participants performed the skill twice. General encouragement, but no verbal feedback on performance was given during or after the tests. All skills were video-recorded and later assessed by one trained assessor who had not administered the tests. The time taken to assess each child was approximately 40 min.

Using the media player classic software, the videos were analyzed to evaluate skills’ criteria. Two Professors in the motor behavior field, with experience in assessing the TGMD-2, carried out a training process on the protocol’s criteria with a master student who did not participate in data assessment. The training process was carried out over 2 weeks and 10% of the videos were randomly analyzed twice by the evaluator, with an interval of 10 days between each evaluation, to determine the intraclass correlation coefficient (ICC). A high agreement for the locomotion score: ICC = 0.93 (95% CI: 0.69–0.98), for the object control score: ICC = 0.98 (95% CI: 0.93–0.99) and for total motor score (MS): ICC: 0.96; (95% CI: 0.82–0.99) were observed. The locomotion and object control scores were based on the presence (one) or absence (zero) of each of the performance criteria. For each subtest, the sum of the raw scores varied from 0 to 48 points.

#### 2.4.4. Executive Function

EF was assessed using Early Years Toolbox—EYT. The “EYT” is a battery of computerized tasks that was developed to assess the EF of children aged 3–5 years-old [34]. The battery consists of five tasks assessed from games in an app designed for iPad. In preschoolers’ immature-brains, IC, working memory, and cognitive flexibility share common processes being challenging to disassociate [3]. Additionally, it is relevant to consider that in early childhood, these components are strongly related to inhibition, both at the representational level or to the maintenance of objectives [35,36,37]. Thus, EF was measured using the Fish and Shark, a typically Go/No Go task. Children were instructed to tap the screen whenever they saw a fish (Go) and not tap the screen when a shark appeared (No Go). Therefore, children must hold their attention and focus on the task but also must inhibit motor responses to the target stimulus [38]. Each stimulus trial remained on the screen for 1500ms, followed by inter stimulus interval of 1000ms. Children completed three blocks of 25 trials (a total of 75 test trials), with 80% Go trials (60 fishes) and 20% No Go trials (15 sharks) that were presented in randomized order. Before beginning the test, each child had the opportunity to hear the app instruction, and to do a full task practice for familiarity. Accuracy of Go, a measure of sustained attention; accuracy of No Go, which is related to inhibition processes; RT of Go, related to speed of response selection were considered, and RT for incorrect responses was set to missing. Previous studies have shown that the Go/No Go tests can activate the entire prefrontal cortex (the brain region considered the basis of support for EF), and it is a more robust task than others to establish EF performance [37,39]. For analysis, one point was assigned for each correct answer, with the score ranging from 0 to 60 points for Go and 0 to 15 points for No Go. This protocol presents satisfactory reliability values with Cronbach’s α = 0.95 [34]. In the current study, the composite reliability value for Go/No Go was 0.78, which is considered an adequate value.

### 2.5. Statistical Procedures

Variables were checked for normality using Kolmogorov–Smirnov tests, and described as categorical or continuous or variables. Chi-squared test was used to calculate the association between proportion of screen time guideline adherence and sex, and Cohen’s d was used to assess effect size among continuous variables [40], defined as small (≥0.2), medium (≥0.5), and large (≥0.8). Statistical significance was accepted, a priori, at *p* < 0.05, and all data were analyzed using SPSS Windows v 20.0 (SPSS, Inc., Chicago, IL, USA).

A network analysis was used to assess the association between adherence to screen time guidelines, FMS, and EF trials, considering children’s age, sex, and BMI. The betweenness, closeness, and strength centrality indicators were reported. Variables with higher betweenness values are more sensitive to changes and may act as a hub, connecting other pairs of variables in the network. A variable with a high closeness value will be quickly affected by changes in any part of the network and may also affect other parts. The strength indicator is essential to understand which variables present the most robust connections in the current network pattern.

The “Fruchterman–Reingold” algorithm was applied. Data were shown in the relative space in which variables with stronger permanent association remain together and with less strongly applied variations repelled one another [41]. To improve the accuracy of the network we used the model “random fields of pair wise Markov”. The algorithm adds a “L1” (regularized neighborhood regression) penalty. The regulation is estimated by a less complete selection and contraction operator (Lasso) that controls the sparse network. The extended Bayesian information criterion (EBIC) to select the Lambda of the regularization parameter was observed. EBIC uses a hyperparameter (y) that determines how much EBIC selects sparse models [42,43]. The y value was determined at 0.25 (range from 0 to 0.50), which is a more parsimonious value in exploratory networks, as proposed in the current study. The network analysis uses least absolute shrinkage and selection operator (LASSO) regularized algorithms to get the precision matrix (weight matrix). When standardized, this matrix represents the associations between the variables in the network. The network is presented in a graph that includes the variables (nodes) and the relations (lines). The full lines represent positive associations and the dashed lines represent negative ones. The thickness and intensity of the lines represent the magnitude of the associations. The qgraph package of RStudio [44] was used to plot the aforementioned figures.

## 3. Results

For screen time, the majority of boys and girls did not comply with WHO recommendations and had similar proportions (x^2^ = 1.87; *p* = 0.172). The boys showed better performance in locomotion, object control, accuracy on Go trial, and reaction time and the girls performed better on accuracy No Go. The effect size was large for all comparisons (Cohen’s *d* > 0.50), except for BMI (Table 1).

The current network standard (Figure 1; Table 2) indicates that compliance with the screen time recommendation is related to the decrease in BMI (−0.52), with accuracy of Go (0.33), decrease in No Go accuracy (−0.26) and also with the improvement of object control skills (0.14). As children get older, there is better compliance with screen time guidelines (0.25) and better accuracy of Go and No Go (0.09 and 0,44, respectively).

Screen time and sex showed the highest strength values, 0.939 and 1.354, respectively, these variables showed the strongest relationships in the network. The variables Ac-Go (2.275), and sex (0.659) presented the highest values of betweenness and the variables Ac-Go (0.714), sex (1.740), and screen time (0.928) have a higher closeness value (Table 3).

## 4. Discussion

This study aimed to assess the associations between compliance with screen time recommendation, EF, and FMS in preschoolers. Although these association have been commonly examined separately, cognition and motor aspects share the same brain basis [11,45] and are affected by screen time. It therefore makes sense to examine these constructs together. Thus, this study adds important information to the literature, accounting for the nonlinear, dynamic, and complex relationship between screen time, EF, and FMS in young children.

The current results showed low compliance with screen time recommendation (less than 10%). This is lower than the percentage observed for preschoolers from Canada (24.4%), Australia (17.3%), Belgium (61%), and Portugal (20.3%), while no differences between sexes has been seen [19,20,21,46]. Possible contextual reasons for screen time patterns has been reported, such as sociodemographic status, single child or siblings, parent’s marital status, parents’ level of awareness regarding the impact of screen time in children’s healthy, and maternal behavioral factors (include age, educational level, pre-pregnancy behaviors, and emotional regulation) [47,48]. These findings highlight the importance of promoting information about screen time guidelines to parents of preschool children.

Several computational measures have been proposed to assess the EF in childhood, based mainly on general accuracy scores. General accuracy of Go/No Go task represents the proportion of correct responses in frequent stimulus and less frequent stimulus [49]. In the present study, the accuracy was computed separately for Go and No-Go trials, and mean RT were calculated for correct Go trials. This analysis allows a more detailed view of the results, once the performance in these variables differ according to age [50,51]. Wiebe et al. [51] showed that young children responded more quickly and accurately on Go trials, but had greater difficulty in inhibiting responding on No-Go trials. Indeed, both RT and accuracy have shown progressive improvement throughout preschool years. In addition, the developmental trajectory of RT, as well as the capacity for attention and inhibition, occurs in different ways throughout early childhood [8,52,53].

The results of the current study also highlighted that girls were more accurate than boys on No-Go trials, reflecting their better inhibition response. Gender differences in emotion expressions, and sensitivity to feedback contribute to differences in inhibitory capacity between sexes [54,55,56]. For example, different neurophysiological responses to negative feedback have been demonstrated, where girls showed more chances of frequent losses and errors, compared to boys [54]. Thus, we may hypothesize that girls’ strategies emphasize avoiding errors commission (No-Go). 

It was also observed that boys performed better in both locomotor and object control skills than girls. This is not surprising as several studies reported that boys better performed total FMS scores, especially on object control skills [57,58]. Nonetheless, both boys and girls should permanently be encouraged to be physically active. Additionally, sex and compliance with screen time recommendations showed the highest strength in the network, highlighting that compliance with screen time is strongly connected with other nodes (age, Ac-Go, and object control score) in the emerged network pattern. As children tend to master in locomotion skills earlier than in object control ones, the variability observed for object control skills may explain its significant results observed, in detriment of locomotion skills. 

The current study also showed that the accuracy of Go, which represents the ability to sustain attention, showed the highest betweenness in the network. This centrality measure indicates that accuracy of Go has the greater connectivity with other nodes in the network, being the most sensitive to potential intervention changes. Indeed, children with greater skills for sustained attention exhibited greater IC, showing a close relationship between the development of attention capacity and improvements in EF during childhood [55,59,60]. Garon et al. [8] have previously reported that maturing attention span forms a basis for the development of EF skills during preschool and, in fact, can be a common source of variation underlying several EF skills. 

Moreover, sex, compliance with screen time recommendations, and accuracy of Go showed the greatest closeness. These variables are responsible for spreading the effect of interventions more quickly on all other variables present in the network. Improving FMS, maintaining or improving BMI, and improving RT in children initially depends on improving screen time (namely compliance with guidelines recommendation), and accuracy of Go. As no difference in compliance with screen time has been seen between boys and girls, the strength of this variable in the network is related to similar opportunities for boys and girls. 

It is also important to mention that environmental factors, such as screen time, could negatively influence both the performance in FMS and in EF tasks in early childhood [27,61,62]. The low amount of time children are engaged in physical practice enables motor experiences [63]. Additionally, previous studies have demonstrated the relationship between screen time with worse inattention, self-regulation problems, and emotional disorders [64,65]. Furthermore, Sigman [61] report that dopamine, a hormone related to the ability to pay attention, is produced in response to screen novelty, and excessive screen time may lead to long-term changes in the reward circuitry that resemble the effects of substance dependence. Moreover, the critical role of dopamine in both motor and cognitive functions [11] provide reasons why these variables should be considered indissociably. Nonetheless, the impact of screen exposure on EF in preschoolers is controversial. Indeed, screen time may be either passive or active [66]. Passive screen time refers to the viewing of screen content that requires little or no interaction from the user, as in television viewing, which has been associated to language and cognitive impairment in young children [67,68,69]. Although the type of screen time has not been reported, in the current study, the assessed children spend a great amount of time in passive screen time, as television viewing is cultural in the Brazilian context, and many families have no access to internet, similar to other Latino countries [70,71]. Moreover, the variety of EF tests used in the studies makes it difficult to extrapolate results. For example, several studies have used measures involving multiple aspects of EF as self-regulation, which makes it difficult to investigate specific components of EF and screen time exposure [47,72,73]. In the present study, the EF was assessed using a Go/No Go test, which allows to obtain data regarding the performance of attention, processing speed, and IC of children.

Accuracy of Go, sex, and compliance with screen time recommendations showed greater closeness in the network. Evidence suggests that, regardless of gender, few preschoolers meet the recommendations of screen time [19,46]. However, there seems to be a consensus regarding the implementation of strategies aimed at promoting children’s compliance with screen time recommendations, in order to positively influence children’s cognitive health [19,74]. Thus, from the findings of the present study, it can be speculated that the change in the screen time may influence the attention capacity of children, which, indirectly, will have reflections on other components such as RT and inhibitory capacity. 

In addition, recent findings demonstrate that children with lower screen time demonstrated better EF, and children who engaged in more PA demonstrated better gross motor skills [75]. In this sense, strategies to reduce screen time and promote activities, especially those that enhance mastery in motor skills, are important for a child’s daily life. From this perspective, the inclusion of adequate and planned physical activities for the age group can contribute not only to motor gain, but also cognitive, since interesting tasks for the child become an object of attention [55].

The strengths of this study include the computerized Go/No Go assessment task that improves the standardization and sensitivity of EF performance; as far as we know, this is the first study to report the relationship between compliance with screen time recommendations, EF, and FMS in preschoolers through a network perspective. However, some limitations should be highlighted. The lack of information regarding the type of screen time (passive or active), the content of the screen offered to the children (educational or entertainment media), as well as the parent-reported screen time are notable limitations. However, it is also worth mentioning that there is no objective validated method of assessing screen time in preschool-aged children. Moreover, the assessed children spend 10 h per day in preschool settings, where electronic devices are forbidden, and screen time is restricted to home time, likely under parent supervision. 

Further, the present study comprises a specific low-income sample. It is known that children from low-income families, and living in socially vulnerable environments [16,76], are more likely to display reduced cognition. Accordingly, the present results should not be extrapolated to the general population. Nonetheless, this study provides novel insight for a specific-targeting population that could benefit from the current evidence. Additionally, information concerning parents’ behaviors, especially the maternal data, and children’s time spent on different types of structured and unstructured PA should be included in future studies. 

Thus, the present study shows that in the emerged network, independent of sex, the compliance with screen time recommendation affects the accuracy on the Go task, in preschoolers. The relationship between these components affects other variables present in the network, such as motor abilities and tasks involved in inhibitory control. Future studies should focus on interventions that reduce screen exposure in younger children.

## Figures and Tables

**Figure 1 ijerph-17-08861-f001:**
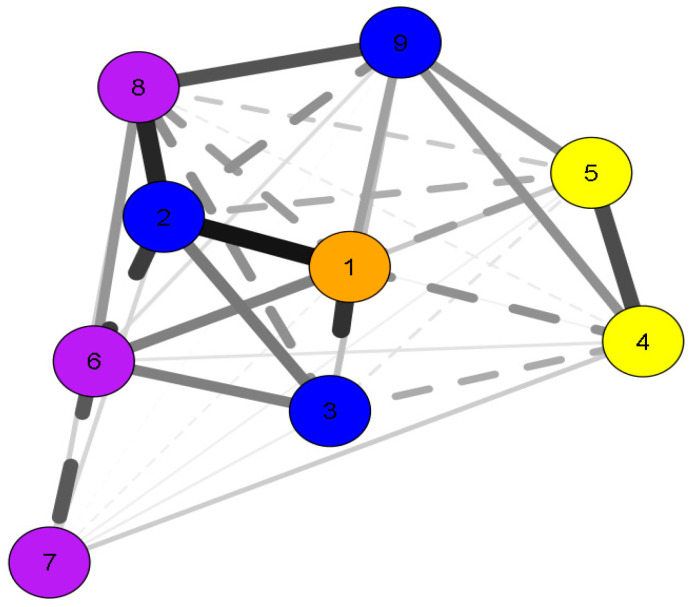
Network associations between compliance with screen time recommendation, sex, body mass index, fundamental motor skills, executive functions, and age. Orange node = compliance with screen time recommendation; blue nodes = anagraphics and anthropometrics; yellow nodes = fundamental motor skills; purple nodes = executive function trials; 1 = compliance with screen time recommendation; 2 = sex; 3 = body mass index; 4 = locomotion score; 5 = object control score; 6 = accuracy of Go; 7 = reaction time of Go; 8 = accuracy of No Go; 9 = age.

**Table 1 ijerph-17-08861-t001:** Sample characteristics and differences.

Variables	N (%)			
	Boys (*n* = 24)	Girls (*n* = 18)	Total (*n* = 42)	x^2^ (*p*)
Screen time (min/day)				
Compliant	1 (4.2)	3 (16.7)	4 (9.6)	1.87(0.172)
Noncompliant	23 (95.8)	15 (83.3)	38 (90.4)	
				*p* (Cohen’s *d*)
Age (years)	3.91 ± 0.77	3.55 ± 0.61	3.76 ± 0.72	0.112 (0.50)
BMI	15.43 ± 1.14	15.51 ± 2.04	15.47 ± 1.57	0.876 (−0.04)
Locomotor	26.33 ± 8.72	21.0 ± 7.77	24.05 ± 8.65	0.047 (0.64)
Object control	22.04 ± 7.45	17.11 ± 7.12	19.93 ± 7.63	0.037 (0.67)
Ac-Go (score)	53.25 ± 5.62	47.22 ± 11.18	50.67 ± 8.87	0.027 (0.71)
RT-Go (ms)	285 ± 0.32	301 ± 0.25	292 ± 0.30	0.083 (−0.55)
Ac-No Go (score)	9.37 ± 3.76	11.83 ± 2.55	10.42 ± 3.48	0.022 (−0.74)

BMI: body mass index; Ac-Go: accuracy on Go; RT-Go: reaction time on Go; Ac-NoGo: accuracy on No Go.

**Table 2 ijerph-17-08861-t002:** Weights matrix.

Variable	Network								
	1	2	3	4	5	6	7	8	9
Screen time (1)	0.00								
Sex (2)	0.58	0.00							
BMI (3)	−0.52	0.35	0.00						
Locomotor (4)	−0.29	0.03	−0.22	0.00					
Object control (5)	0.14	−0.22	−0.07	0.45	0.00				
Ac-Go (6)	0.33	−0.49	0.33	0.08	−0.25	0.00			
RT-Go (7)	−0.04	0.11	0.05	0.13	0.04	−0.42	0.00		
Ac-No Go (8)	−0.26	0.54	−0.29	−0.05	−0.14	0.27	0.12	0.00	
Age (9)	0.25	−0.29	0.17	0.29	0.28	0.09	−0.01	0.44	0.00

1 = compliance with screen time recommendation; 2 = sex; 3 = body mass index; 4 = locomotion score; 5 = object control score; 6 = accuracy of Go; 7 = reaction time of Go; 8 = accuracy of No Go; 9 = age.

**Table 3 ijerph-17-08861-t003:** Centrality measures.

Variable	Betweenness	Closeness	Strength
Age	0.120	−0.501	−0.214
Screen time	−0.419	0.928	0.939
Sex	0.659	1.740	1.354
BMI	0.120	−0.127	0.149
Locomotion	−0.419	−0.702	−0.734
Object control	0.242	−0.781	−0.633
Ac-Go	2.275	0.714	0.668
RT-Go	−0.958	−1.479	−1.919
Ac-No Go	−0.419	0.210	0.390

BMI: body mass index; Ac-Go: accuracy on Go; RT-Go: reaction time on Go; Ac-No Go: accuracy on No Go.

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
