# Peer review of "A Network Perspective on the Relationship between Screen Time, Executive Function, and Fundamental Motor Skills among Preschoolers"

_ijerph, 2020, doi:10.3390/ijerph17238861_

Round 1
Reviewer 1 Report
This article focuses on the relationship between screen time, executive function, and fundamental motor skills of preschoolers in Brazil is very interesting. The topic is important and the paper is well organized with detailed information on backgrounds, data collection, and analysis, as well as results and discussion. However as the explanation of the main purpose of identifying “networks” is limited, several parts of the article remain unclear. It could be accepted for publishing, however, revision is needed. I would like to give the following questions and suggestions for the authors to consider during their revision:
1) This paper attempted to identify the relationship between “screen time”, “executive function”, and “the fundamental motor skills” from the perspective of "network", but failed to explain the details of the “science of network” perspective, and how the authors would use it in the current study. Information should be added to explain how this perspective is validated.
2) following with the previous question, from the perspective of "network", it is difficult to conclude that “independent of sex, screen exposure affects the accuracy on Go task, and these components affect the variables in the network, as motor abilities and tasks involved in inhibitory control”. There are many confounding variables that should be considered. To give a little: targeted children’s time spend on sports and sport-like activities in both school and holidays; and the attribution of different types of physical activities in the targeted children on daily basis, are all important variables, however, have not been considered in this article. Further explanations should be made.
3) About participants, as the majority of the children come from low income and low education status families, would these variables also influence the results?
4) About screen time, instead of observing targeted children directly, parents were asked. Limitations of this approach should be considered, such as the absence of parents during the daytime in the life of their children? How the accuracy of parents’ documentation of screen time of children has been confirmed?
5) The method used for accessing EF, authors utilized a game on iPad to collect relevant data. Would this method validate in this study? Would the screen game experience among children influence the data? It might be possible that children who spent more time on-screen games are more familiar with the operating of any games on the iPad?
Author Response
To reviewer 1,
Thank you for giving us the possibility to submit a revised version of the manuscript entitled “A Network Perspective On The Relationship Between Screen Time, Executive Function, And Fundamental Motor Skills Among Preschoolers” (IJERPH- 987976) to the International Journal of Environmental Research and Public Health. The authors thank the reviewers for the thoughtful comments and suggestions on our manuscript. We have considered all of the suggestions and have incorporated them into the revised manuscript and we believe our manuscript is stronger as a result of these modifications.
________________________________________________________________
Comments Reviewer # 1
_______________________________________________________________
This article focuses on the relationship between screen time, executive function, and fundamental motor skills of preschoolers in Brazil is very interesting. The topic is important and the paper is well organized with detailed information on backgrounds, data collection, and analysis, as well as results and discussion. However as the explanation of the main purpose of identifying “networks” is limited, several parts of the article remain unclear. It could be accepted for publishing, however, revision is needed. I would like to give the following questions and suggestions for the authors to consider during their revision:
______________
Comment #1
This paper attempted to identify the relationship between “screen time”, “executive function”, and “the fundamental motor skills” from the perspective of "network", but failed to explain the details of the “science of network” perspective, and how the authors would use it in the current study. Information should be added to explain how this perspective is validated.
Authors´ response #1
The comments are highly appreciated. We´ve added a paragraph in introduction section justifying the network approach used (see below), beside a detailed explanation along the methods, and discussion sections to support our point of view.
“The network science, as a way of measuring complex systems, provides a topological structure of network that can be useful to understand non-linear relationships and is very sensitive to initial conditions. In addition, the role of each variable in the network can be better understood from the centrality measures, used to identify critical areas in the network that may be optimize though intervention processes. The network science is an emerging research area used to understand and to measure complex systems.”
________________
Comment #2
Following with the previous question, from the perspective of "network", it is difficult to conclude that “independent of sex, screen exposure affects the accuracy on Go task, and these components affect the variables in the network, as motor abilities and tasks involved in inhibitory control”. There are many confounding variables that should be considered. To give a little: targeted children’s time spend on sports and sport-like activities in both school and holidays; and the attribution of different types of physical activities in the targeted children on daily basis, are all important variables, however, have not been considered in this article. Further explanations should be made.
Authors´ response #2
We totally agree with reviewer comment. Indeed, in dynamic systems, every small change in sub-dynamic systems may change the entire network. Thus, we´ve rewritten the conclusion sentence as follow:
“Thus, the present study shows that in the emerged network, independent of sex, the compliance with screen time recommendation affects the accuracy on Go task, in preschoolers.”
Additionally, we´ve inserted a sentence including directions for future studies as follow:
“Additionally, information concerning parents' behaviours, especially the maternal data, and children’s time spend on different types of structured and unstructured PA should be included in future studies.”
__________________
Comment #3
About participants, as the majority of the children come from low income and low education status families, would these variables also influence the results?
Authors´ response #3
Thank you for your comment. The participants of the present study are from low-income families, and live in vulnerable areas. In this sense, between-subjects variability for this variable is limited. Nonetheless, we totally agree with reviewer concern, and added the following sentence in the limitations paragraph:
“Further, the present study comprises a specific low-income sample. It is known that children from low-income families and living in socially vulnerable situation, are more likely to show reduced cognition, and results should not be extrapolated to the general population. Nonetheless, this study comprises a specific-targeting population that could benefit from the current evidences provided.”
_______________
Comment #4
About screen time, instead of observing targeted children directly, parents were asked. Limitations of this approach should be considered, such as the absence of parents during the daytime in the life of their children? How the accuracy of parents’ documentation of screen time of children has been confirmed?
Authors´ response #4
We appreciate reviewer comment and inserted the following sentence in discussion section:
“However, some limitations should be highlighted. The lack of information about the type of screen time (passive or active), the content of the screen offered to the children (educational or entertainment media), as well as the parent-reported screen time are notable limitations, though it is also worth mentioning that there is no validated objective method of assessing screen time in preschool-aged children. Moreover, those children spend 10 hours per day at preschools setting, where electronic devices are forbidden, and screen time is restricted to home time, probably under parents ‘supervision.”
____________________
Comment #5
The method used for accessing EF, authors utilized a game on iPad to collect relevant data. Would this method validate in this study? Would the screen game experience among children influence the data? It might be possible that children who spent more time on-screen games are more familiar with the operating of any games on the iPad?
Authors´ response #5
Thank you for your comment. We understand reviewer´s concern about child´s experience with electronic devices and how this could influence our results. Unfortunately, we have no data on children´s experience with these devices. Nonetheless, executive demand is task-specific, and this protocol shows satisfactory reliability values for assessing young children´s executive function. Moreover, during the assessment, each child had the opportunity of doing a full task practice, which could minimize the bias risk. It is also worthy to mention that the task consists only of touching the screen, which we believe does not necessarily require great manual screen skills. Thus, we´ve added the following sentence in the methods section, and hope reviewer agree with our point of view:
“Before beginning the test, each child had the opportunity to hear the app instruction, and to do a full task practice for familiarity.”
Reviewer 2 Report
The manuscript by Martins and colleagues investigates an impacting issue of contemporary society, i.e., screen exposure in preschoolers. In this overall well-written work, the authors appropriately perform a dynamic and non-linear analysis using the science of networks to assess the association between screen time, motor skills and executive functions in 3-to-5 years-old children. This is an innovative way to investigate the complex connectivity among motor and cognitive functions and the environment.
However, the methodology used is not adequately described. In my opinion, this is the main drawback of the present work. Moreover, the results can be presented in a clearer way, by correcting some inaccuracies.
In particular:
- Methods – a Statistical Analysis section should be added.
Here, the tests performed to compare groups (boys vs. girls) with regard to several variables (e.g., screen time, age, BMI, EF, motor skills, etc) should be described. Moreover, why only Cohen’s d of these comparisons is indicated? In my opinion, a t-test and the p values should also be performed and presented.
Furthermore, the methodology used for network analysis should be briefly explained. Definitions of the centrality measures presented (betweenness, closeness, strength) should be provided in order to help the reader to follow the manuscript.
- Results – some passages may present inaccuracies or missing information that make them hard to follow.
When presenting the main result of network analysis (lines 180-184), the authors state that “compliance with the screen time guidelines is related to” different other nodes. However, in Figure 1 and later on in the text, the variable is referred as “screen time”. It is crucial to clarify this point, because if we consider compliance with screen time we assume that the variable indicate few screen exposure (≤ 1 or 2 hours/day accordingly with age). Conversely, screen time indicate the raw amount of screen exposure. Therefore, the positive/negative weights change in their interpretation.
In figure 1, should the connection 1 -> 5 be green (value 0.14)?
As previously suggested, a previous explanation on the basic analyses and measures used could help to follow the results section.
Minor comments:
Line 31 (and 240): “sensitivity” is used as an adjective, but is a noun. Should be “most sensitive”? Moreover and more important, it is not clear if being the most sensitive is an interpretation of the results obtained or is a direct consequence of the outcome.
Line 85: the repetition of the word “used” could be avoided (maybe using “and” instead of the second)
Line 91: PA was not previously defined. I guess it stand for physical activity, but should be indicated in the text.
Line 111: TGMD-2 is defined at the second use in the text. Please move at first use.
Line 153: “YET” should be “EYT”. Moreover, a “.” is missing.
Figure 1. Please indicate in the caption a legend for nodes’ colors (e.g., blue=anagraphics and anthropometrics; purple=EF; yellow=FMS) and for connections’ color and thickness. Using the same colors than Table 2 could be helpful.
Line 182-184: how was this detected? What kind of analysis revealed this? If it is from the weights between factors presented in Table 2, I can find the relation between age and screen time (0.25), but not between age and accuracy of Go (in the table, I can see 0.09 and not 0.33 as reported in the text).
Line 193: separate sentences with “.” or “;”.
Line 194: should be “the strongest relationship”?
Line 193-196: please refer to Table 3 in the text (which, in turn, misses numeration). Again, a previous explanation on analysis outcomes (betweenness, closeness, strength) would be useful.
Line 204: remove the word “when” or the following comma
Line 212: remove “report”
Line 284: what PE stand for?
Author Response
To reviewer 2,
Thank you for giving us the possibility to submit a revised version of the manuscript entitled “A Network Perspective On The Relationship Between Screen Time, Executive Function, And Fundamental Motor Skills Among Preschoolers” (IJERPH- 987976) to the International Journal of Environmental Research and Public Health. The authors thank the reviewers for the thoughtful comments and suggestions on our manuscript. We have considered all of the suggestions and have incorporated them into the revised manuscript and we believe our manuscript is stronger as a result of these modifications.
____________________________________________________________________________
Reviewer # 2
____________________________________________________________________________
The manuscript by Martins and colleagues investigates an impacting issue of contemporary society, i.e., screen exposure in preschoolers. In this overall well-written work, the authors appropriately perform a dynamic and non-linear analysis using the science of networks to assess the association between screen time, motor skills and executive functions in 3-to-5 years-old children. This is an innovative way to investigate the complex connectivity among motor and cognitive functions and the environment. However, the methodology used is not adequately described. In my opinion, this is the main drawback of the present work. Moreover, the results can be presented in a clearer way, by correcting some inaccuracies.
______________
Comment #1
Methods – a Statistical Analysis section should be added.
Here, the tests performed to compare groups (boys vs. girls) with regard to several variables (e.g., screen time, age, BMI, EF, motor skills, etc) should be described. Moreover, why only Cohen’s d of these comparisons is indicated? In my opinion, a t-test and the p values should also be performed and presented.
Furthermore, the methodology used for network analysis should be briefly explained. Definitions of the centrality measures presented (betweenness, closeness, strength) should be provided in order to help the reader to follow the manuscript.
Authors´ response #1
We totally agree with reviewer comment. We have inserted the comparison between sexes, and a topic detailing the statistical procedures used, and hope it is well-explained for readers (please see table 1 and item 2.5).
“2.5. Statistical Procedures
Variables were checked for normality using Kolmogorov–Smirnov tests, and described as categorical or continuous or variables.]. Chi-squared Test was used to calculate the association between proportion of screen time guideline adherence and sex, and Cohen’s d was used to assess effect size among continuous variables, defined as small (≥0.2), medium (≥0.5), and large (≥0.8). Statistical significance was accepted, a priori, at p<0,05, and all data were analyzed using SPSS Windows v 20.0 (SPSS, Inc., Chicago, IL, USA).
A network analysis was used to assess the association between adherence to screen time guidelines, FMS, and EF trials, considering children’s age, sex and BMI. The betweenness, closeness and strength centrality indicators were reported. Variables with higher betweenness values are more sensitive to changes and may act as a hub, connecting other pairs of variables in the network. A variable with a high closeness value will be quickly affected by changes in any part of the network and may also affect other parts. The strength indicator is essential to understand which variables present the most robust connections in the current network pattern.
The “Fruchterman–Reingold” algorithm was applied. Data were shown in the relative space in which variables with stronger permanent statistics together and with less strongly applied variations repelled one another. To improve the accuracy of the network we used the model “random fields of pair wise Markov”. The algorithm adds a “L1” (regularized neighborhood regression) penalty. The regulation is estimated by a less complete selection and contraction operator (Lasso) that controls the sparse network. The Extended Bayesian Information Criterion (EBIC) to select the Lambda of the regularization parameter was observed. EBIC uses a hyperparameter (y) that determines how much EBIC selects sparse models. The y value was determined at 0.25 (range from 0 to 0.50), which is a more parsimonious value in exploratory networks, as proposed in the current study. The network analysis uses Least absolute shrinkage and selection operator (LASSO) regularized algorithms to get the precision matrix (weight matrix). When standardized, this matrix represents the associations between the variables in the network. The network is presented in a graph that includes the variables (nodes) and the relations (lines). The full lines represent positive associations and the dashed lines represent negative ones. The thickness and intensity of the lines represent the magnitude of the associations. The qgraph package of RStudio was used to plot the afformentioned figures.”
______________
Comment #2
Results – some passages may present inaccuracies or missing information that make them hard to follow.
When presenting the main result of network analysis (lines 180-184), the authors state that “compliance with the screen time guidelines is related to” different other nodes. However, in Figure 1 and later on in the text, the variable is referred as “screen time”. It is crucial to clarify this point, because if we consider compliance with screen time we assume that the variable indicate few screen exposure (≤ 1 or 2 hours/day accordingly with age). Conversely, screen time indicate the raw amount of screen exposure. Therefore, the positive/negative weights change in their interpretation.
In figure 1, should the connection 1 -> 5 be green (value 0.14)?
As previously suggested, a previous explanation on the basic analyses and measures used could help to follow the results section.
Authors´ response #2
We are thankful for reviewer´s comment and we´ve corrected all the inaccuracies, clarifying that the screen time was analyzed as a categorical variable (compliant / non-compliant), and adding a previous explanation regarding analysis procedures. Concerning the connection between nodes 1 and 5, actually we´ve had overlapping lines (nodes 5 and 6; and nodes 1 and 5). To a better visualization, we´ve changed the figure and added full and dashed lines, hopping it is easier to comprehend.
______________
Comment #3
Minor comments:
Line 31 (and 240): “sensitivity” is used as an adjective, but is a noun. Should be “most sensitive”? Moreover and more important, it is not clear if being the most sensitive is an interpretation of the results obtained or is a direct consequence of the outcome.
Line 85: the repetition of the word “used” could be avoided (maybe using “and” instead of the second)
Line 91: PA was not previously defined. I guess it stand for physical activity, but should be indicated in the text.
Line 111: TGMD-2 is defined at the second use in the text. Please move at first use.
Line 153: “YET” should be “EYT”. Moreover, a “.” is missing.
Figure 1. Please indicate in the caption a legend for nodes’ colors (e.g., blue=anagraphics and anthropometrics; purple=EF; yellow=FMS) and for connections’ color and thickness. Using the same colors than Table 2 could be helpful.
Line 182-184: how was this detected? What kind of analysis revealed this? If it is from the weights between factors presented in Table 2, I can find the relation between age and screen time (0.25), but not between age and accuracy of Go (in the table, I can see 0.09 and not 0.33 as reported in the text).
Line 193: separate sentences with “.” or “;”.
Line 194: should be “the strongest relationship”?
Line 193-196: please refer to Table 3 in the text (which, in turn, misses numeration). Again, a previous explanation on analysis outcomes (betweenness, closeness, strength) would be useful.
Line 204: remove the word “when” or the following comma
Line 212: remove “report”
Line 284: what PE stand for?
Authors´ response #3
The authors appreciate reviewer corrections and added all of them in the text.

Round 2
Reviewer 1 Report
The manuscript has been well revised with detailed information and it can be accepted for publishing. However, minor adjustiment of the figure colors to make them easier to read and a careful recheck of the language and the references are necessary. About figure color: e.g., Fig. 1, numbers in the blue nodes are not easy to read; the different degrees of red and blue colors in Table 2 are also confusing.